# KALMAN FILTER FOR ONLINE CLASSIFICATION OF NON-STATIONARY DATA

**Michalis K. Titsias**[*]
Google DeepMind
mtitsias@google.com

**Alexandre Galashov**[*]
Google DeepMind
agalashov@google.com

**Amal Rannen-Triki**
Google DeepMind
arannen@google.com

**Razvan Pascanu**
Google DeepMind
razp@google.com

**Yee Whye Teh**
Google DeepMind
ywteh@google.com

**Jörg Bornschein**
Google DeepMind
bornschein@google.com

## ABSTRACT

In Online Continual Learning (OCL) a learning system receives a stream of data and sequentially performs prediction and training steps. Key challenges in OCL include automatic adaptation to the specific non-stationary structure of the data and maintaining appropriate predictive uncertainty. To address these challenges we introduce a probabilistic Bayesian online learning approach that utilizes a (possibly pretrained) neural representation and a state space model over the linear predictor weights. Non-stationarity in the linear predictor weights is modelled using a "parameter drift" transition density, parametrized by a coefficient that quantifies forgetting. Inference in the model is implemented with efficient Kalman filter recursions which track the posterior distribution over the linear weights, while online SGD updates over the transition dynamics coefficient allow for adaptation to the non-stationarity observed in the data. While the framework is developed assuming a linear Gaussian model, we extend it to deal with classification problems and for fine-tuning the deep learning representation. In a set of experiments in multi-class classification using data sets such as CIFAR-100 and CLOC we demonstrate the model's predictive ability and its flexibility in capturing non-stationarity.

## 1 INTRODUCTION

Continual Learning (e.g. Hadsell et al., 2020; Parisi et al., 2019) is an open problem that has been receiving increasing attention in recent years. It aims to provide answers on how to train and use models in non-stationary scenarios. A multitude of different and sometimes conflicting desiderata have been considered for continual learning, including forward transfer, backward transfer, avoiding forgetting and maintaining plasticity. Training and evaluation protocols highlight different constraints such as limited memory to store examples, limited model size or computational constraints.

In this work we focus on the Online Learning (OL) (Shalev-Shwartz, 2012; Hazan, 2017) scenario, where a learner receives a sequence of inputs $x_n$ and targets $y_n$. At each time-step, the model observes $x_n$, makes a prediction and then receives the associated loss and ground truth target for learning. Within the deep learning community this scenario has been referred to as *Online Continual Learning* (OCL) (Cai et al., 2021; Ghunaim et al., 2023), with the focus shifted more towards obtaining the best empirical performance on some given data instead of bounding the worst-case regret. Note that in OL/OCL each observation $(x_n, y_n)$ is first used for evaluation before it is used for training. No separate evaluation-sets are required and this objective naturally supports task-agnostic and non-stationary scenarios. Depending on the nature of the data stream, plasticity, not-forgetting and sample-efficiency all play a crucial role during learning. When considering the cumulative next-step log-loss under this protocol, it directly corresponds to the prequential description length and is thus a theoretically well motivated evaluation metric for non-stationary scenarios under the Minimum Description Length principle (Grunwald, 2004; Blier and Ollivier, 2018; Bornschein et al., 2022b).

---

[*]Joint first authorship

In this paper, we propose a new method based on Kalman filters which explicitly takes into account non-stationaries in the data stream. It assumes a prior Markov model over the linear predictor weights that uses a "parameter drift" transition density parametrized by a coefficient that quantifies forgetting. The prior model is then combined with observations using online Bayesian updates, implemented by computationally fast Kalman filter recursions, that track the posterior distribution over the weights of the linear predictor as the data distribution changes over time. These Bayesian updates are also combined with online SGD updates over the forgetting coefficient, thus allowing for more flexible adaptation to non-stationarity. While the method is developed assuming the tractable linear Gaussian case, we also extend it to classification and for updating deep learning representations.

We follow the trend introduced by large scale pretrained or foundation models (Bommasani et al., 2021) of separating the representation learning from the readout classifier. Foundation models provide generic representations applicable across a multitude of tasks and have been argued to offer stable representations in continual learning, shifting the focus to handling non-stationarity with the convex readout layer. This separation of concerns facilitates a more natural integration of non-neural solutions to address continual or online learning challenges. Following this perspective, we investigate the performance of Kalman filter on top of a frozen pretrained representation. We demonstrate that simultaneously online learning of the representation while performing Kalman filter updates leads to stable learning and provide strong results on online continual learning benchmarks.

Our contributions can be summarised as follows: (i) We propose a Kalman filter based approach which models the parameters drift in the last layer of the Neural Network (NN) and derive efficient Bayesian inference updates. (ii) We introduce an online learning algorithm to learn the drift parameters online therefore adjusting the last layer linear dynamics to the non-stationarity of the data. (iii) We demonstrate that this approach can work with pre-trained features as well as when it is trained from scratch, allowing for training NN representations online. (iv) We demonstrate that the approach achieves superior performance on large-scale benchmarks against existing baselines.

## 2 ONLINE LEARNING WITH KALMAN FILTER

We first describe the method in univariate regression and discuss in Section 2.1 how we adapt it to classification. We assume a stream of data arriving sequentially so that at step $n$ we receive $(x_n, y_n)$ where $x_n \in \mathbb{R}^d$ is the input vector and $y_n \in \mathbb{R}$ is the target. Our objective is to introduce an online learning model that captures non-stationarities and learns a good predictor for future data. Our model consists of two parts. The fist part is a neural network that outputs a representation $\phi(x; \theta) \in \mathbb{R}^m$ so that $\phi_n := \phi(x_n; \theta)$ is the feature vector of the $n$-th data point. $\theta$ is a set of parameters that could be fixed, if the feature extractor is a *pretrained* network, or learnable, when it is either fine-tuned or learnt from scratch. Given $\phi_n$ the output $y_n$ is modelled by a Gaussian likelihood $p(y_n|w_n) = \mathcal{N}(y_n|w_n^\top \phi_n, \sigma^2)$, where the $m$-dimensional vector of regression coefficients $w_n$ depends on time index $n$, i.e. $w_n$ can change with time in order to model distributional changes.

The second part of the model is a Gaussian Markov prior process on how $w_n$ can change over time, which models *parameter drift* based on the following dynamics,

$$p(w_0) = \mathcal{N}(w_0|0, \sigma_w^2 I), \qquad \text{Initial parameter} \qquad (1)$$

$$p(w_n|w_{n-1}) = \mathcal{N}(w_n|\gamma_n w_{n-1}, (1 - \gamma_n^2)\sigma_w^2 I), \quad n \geq 1. \quad \text{Parameter drift} \qquad (2)$$

The time-dependent parameter $\gamma_n$ takes values in $[0, 1]$ and quantifies the *memory or forgetting* of the process. For example, when $\gamma_n = 1$ then $w_n = w_{n-1}$, which means that at time step $n$ the model re-uses (or copies forward) the parameter $w_{n-1}$ from the previous step. Such extreme case is suitable when there is no distributional change at time $n$. In the other extreme case, when $\gamma_n = 0$ the parameter $w_n$ is fully refreshed, i.e. reset to the prior $\mathcal{N}(0, \sigma_w^2 I)$, which implies a sharp change in the distribution. Similarly, intermediate values of $\gamma_n \in (0, 1)$ can model more smooth or gradual changes. Flexible learning of $\gamma_n$ through time will be a key element of our method that we describe in Section 2.1.1 for the classification problem. Furthermore, it is worth noting that the transition model in equation 2 is invariant to the prior, so that when $w_{n-1} \sim \mathcal{N}(0, \sigma_w^2 I)$ the next state $w_n = \gamma_n w_{n-1} + \sqrt{1 - \gamma_n^2}\epsilon$ ($\epsilon \sim \mathcal{N}(0, I)$) also has marginal distribution $\mathcal{N}(0, \sigma_w^2 I)$. Having specified the observation model and the transition model over the parameters $w_n$ we can write the full joint density up to the $n$-th observation as

$$p(w_0)\prod_{i=1}^{n} p(y_i|w_i)p(w_i|w_{i-1}) = \mathcal{N}(w_0|0, \sigma_w^2 I)\prod_{i=1}^{n}\mathcal{N}(y_i|w_i^\top \phi_i, \sigma^2)\mathcal{N}(w_i|\gamma_i w_{i-1}, (1 - \gamma_i^2)\sigma_w^2 I).$$

This is a linear Gaussian state space model where exact online Bayesian inference over $w_n$ can be solved with standard Kalman filter prediction and update steps to compute online the Bayesian posterior over $w_n$ (Särkkä, 2013). The prediction step requires computing the posterior over $w_n$ given all past data up to time $n-1$ (and excluding the current $n$-th observation), which is a Gaussian density $p(w_n|y_{1:n-1}) = \mathcal{N}(w_n|m_n^-, A_n^-)$ with parameters

$$m_n^- = \gamma_n m_{n-1}, \quad A_n^- = \gamma_n^2 A_{n-1} + (1 - \gamma_n^2)\sigma_w^2 I. \tag{3}$$

The update step finds the updated Gaussian posterior $p(w_n|y_{1:n}) = \mathcal{N}(w_n|m_n, A_n)$ by modifying the mean vector $m_n$ and the covariance matrix $A_n$ to incorporate the information coming from the most recent observation $(x_n, y_n)$ according to

$$m_n = m_n^- + \frac{A_n^- \phi_n}{\sigma^2 + \phi_n^\top A_n^- \phi_n}(y_n - \phi_n^\top m_n^-), \quad A_n = A_n^- - \frac{A_n^- \phi_n \phi_n^\top A_n^-}{\sigma^2 + \phi_n^\top A_n^- \phi_n}. \tag{4}$$

The initial conditions for the recursions are $m_0 = 0$ and $A_0 = \sigma_w^2 I$. The complexity is dominated by the operations $s_n = A_n^- \phi_n$ and $s_n s_n^\top$ which are $O(m^2)$ making the Kalman recursions very efficient. Further, if we remove the stochasticity from the transition dynamics by setting $\gamma_n = 1$ for any $n$, the above Kalman recursions reduce to online Bayesian linear regression as detailed in Appendix A.

**Prediction.** As the online model sequentially receives observations and updates its Bayesian posterior distribution over $w_n$, it can also perform next step predictions. Suppose that after $n-1$ steps the model has observed the data $(x_i, y_i)_{i=1}^{n-1}$ and computed the posterior $p(w_n|y_{1:n-1})$. Then for the next input data $x_n$ the model can predict its output $y_n$ based on the Bayesian predictive density

$$p(y_n|y_{1:n-1}) = \int p(y_n|w_n)p(w_n|y_{1:n-1})dw_n = \mathcal{N}(y_n|\phi_n^\top m_n^-, \phi_n^\top A_n^- \phi_n + \sigma^2), \tag{5}$$

which is analytic for this Gaussian regression case, while for the classification case will require approximate inference and Monte Carlo sampling as detailed in Section 2.1.1.

## 2.1 APPLICATION TO CLASSIFICATION

We now apply the above online learning model to multi-class classification problems. Suppose a classification problem with $K$ classes, where the label $y_n$ is encoded as $K$-dimensional one-hot vector, i.e. $y_n \in \{0, 1\}^K$, $\sum_{k=1}^K y_{n,k} = 1$. A suitable observation model for classification is the standard softmax likelihood $p(y_{n,k} = 1|W_n) = \frac{e^{w_{n,k}^\top \phi_n}}{\sum_{j=1}^K e^{w_{n,j}^\top \phi_n}}$, where $W_n = (w_{n,1}, \ldots, w_{n,K})$ is a $m \times K$ matrix storing the parameters $W_n$ which play a similar role to previous regression coefficients. These parameters follow $K$ independent Markov processes so that each $k$-th column $w_{n,k}$ of $W_n$ is independent from the other columns and obeys the transition dynamics from equation 1 and equation 2. However, unlike the regression case, exact online inference over $W_n$ using Kalman recursions is intractable due to the non-Gaussian form of the softmax likelihood. Thus, we need to rely on approximate inference. Next we derive a fast and very easy to implement inference technique that still uses the exact Kalman recursions. This consists of two main components described next.

**(i) Fast and analytic Kalman recursion.** To achieve this we introduce a Gaussian likelihood that (as a means to approximate inference) replaces the softmax likelihood. It has the form $q(y_n|W_n) = \prod_{j=1}^K \mathcal{N}(y_{n,j}|w_{n,j}^\top \phi_n, \sigma^2)$, which explains the elements of the one-hot vector by a Gaussian density, a trick that has been used successfully in the literature e.g. for Gaussian process classification (Rasmussen and Williams, 2006) and meta learning (Patacchiola et al., 2020). With this approximate likelihood the Kalman recursions remain tractable and propagate forward an approximate predictive posterior $q(W_n|y_{1:n-1}) = \prod_{k=1}^K \mathcal{N}(w_{n,k}|m_{n,k}^-, A_n^-)$ with mean parameters given by the $m \times K$ matrix $M_n^- = (m_{n,1}^-, \ldots, m_{n,K}^-)$ and covariance parameters given by the $m \times m$ matrix $A_n^-$:

$$M_n^- = \gamma_n M_{n-1}, \quad A_n^- = \gamma_n^2 A_{n-1} + (1 - \gamma_n^2)\sigma_w^2 I. \tag{6}$$

The corresponding updated posterior $q(W_n|y_{1:n}) = \prod_{k=1}^K \mathcal{N}(w_{n,k}|m_{n,k}, A_n)$ has parameters

$$M_n = M_n^- + \frac{A_n^- \phi_n}{\sigma^2 + \phi_n^\top A_n^- \phi_n} \times (y_n^\top - \phi_n^\top M_n^-), \quad A_n = A_n^- - \frac{A_n^- \phi_n \phi_n^\top A_n^-}{\sigma^2 + \phi_n^\top A_n^- \phi_n}. \tag{7}$$

The recursion is initialized at $M_0 = \mathbf{0}$, $A_0 = \sigma_w^2 I$, where $\mathbf{0}$ denotes the $m \times K$ matrix of zeros. A full iteration costs $O(Km + m^2)$ where the $O(m^2)$ operations are the same with the regression case, while the additional $O(Km)$ operations are needed in the update of $M_n$, such as for $\phi_n^\top M_n^-$. If the feature vector size $m$ is larger than the number of classes $K$, the term $O(m^2)$ dominates and the complexity is the same as in univariate regression. The crucial factor to obtain such an efficiency is that the covariances matrices $(A_n^-, A_n)$ are shared among all $K$ classes.[1] The additional memory overhead of the method is $O(Km)$ for storing $M_n$ and $O(m^2)$ for storing the covariance $A_n$.

**(ii) Posteriors combine with softmax for prediction or parameter updating.** We view the above Kalman recursion as an online approximate inference procedure that provides an approximation to the exact posterior distribution. If $p(W_n|y_{1:n-1})$ is the exact intractable predictive posterior (obtained by Bayes' rule with the exact softmax likelihood) then $q(W_n|y_{1:n-1})$ computed from Kalman recursion is an approximation to $p(W_n|y_{1:n-1})$. Subsequently, by following standard approximate Bayesian inference practices, when we wish to predict class probabilities or compute a cross entropy-like loss to optimize parameters, the approximate posterior is combined with the exact softmax through Bayesian averaging and Monte Carlo estimation. In Section 2.1.1, we make use of this to learn online the forgetting coefficient $\gamma_n$. Further in Appendix C, we use the same principle to compute accurate predictive probabilities and fine-tune the representation parameters $\theta$. For this second case we also find it useful to introduce a Bayesian calibration procedure, which optimizes online a calibration parameter and improves the predictive probability estimates; see Appendix C for full details.

**(iii) Online backbone finetuning.** We also allow for finetuning the backbone $\phi(x_n, \theta)$ online; see Appendix G. In this case, the Kalman recursion will not be exact in the sense that it combines features $\phi(x_n, \theta)$ of different $\theta$ values. Nevertheless, we found that such an approach works well in practice.

### 2.1.1 ONLINE UPDATING THE FORGETTING COEFFICIENT $\gamma_n$

An important aspect of our method is the online updating of the forgetting coefficient $\gamma_n$, which is indexed by $n$ to indicate that it can change over time to reflect the data distributional changes. Intuitively, if the predictive probability of the next data point becomes low, this suggests that the Bayesian posterior does not explain this point well and needs refreshment from the prior. Adjusting $\gamma_n$ will move the posterior either closer or further away from the prior as can be seen from equation 6. Instead of using a full Bayesian approach (that will require an extra hierarchical prior) over $\gamma_n$ we will follow a simple empirical Bayes method where we update $\gamma_n$ by online point estimation. For that, we first initialize $\gamma_0$ and then for any subsequent time step $n \geq 1$, where we observe $(x_n, y_n)$, we first copy the previous $\gamma_{n-1}$ value to the new step, i.e. $\gamma_n = \gamma_{n-1}$, and then we apply an SGD update to change $\gamma_n$ by maximizing the log predictive probability. The SGD update is written as

$$\gamma_n \leftarrow \gamma_n + \rho_n \nabla_{\gamma_n} \log p(y_{n,k} = 1|y_{1:n-1}), \tag{8}$$

where we further parametrize $\gamma_n = e^{-0.5\delta_n}$, with $\delta_n \geq 0$, so that the update is applied to $\delta_n$.[2] However, since the log predictive probability is not tractable we consider the approximation

$$\log p(y_{n,k} = 1|y_{1:n-1}) \approx \log \int p(y_{n,k} = 1|W_n) q(W_n|y_{1:n-1}) dW_n, \tag{9}$$

where $p(y_n|W_n)$ is the softmax and $q(W_n|y_{1:n-1}) = \prod_{k=1}^K \mathcal{N}(w_{n,k}|m_{n,k}^-, A_n^-)$ is the Kalman analytic approximate posterior distribution. To estimate this we can use the standard procedure to reparametrize the integral in terms of the $K$-dimensional vector of logits $f_n = W_n^\top \phi_n$ which follows the factorized Gaussian distribution $q(f_n|\mu_n, s_n^2 I)$ with mean $\mu_n = (M_n^-)^\top \phi_n$ and isotropic variance $s_n^2 = \phi_n^\top A_n^- \phi_n$. After this reparametrization, equation 9 leads to the estimate

$$\log p(y_{n,k} = 1|y_{1:n-1}) = \log \int \frac{e^{f_{n,k}}}{\sum_{j=1}^K e^{f_{n,j}}} q(f_n|\mu_n, s_n^2 I) df_n \approx \log \frac{1}{S} \sum_{s=1}^S \frac{e^{\mu_{n,k} + s_n \epsilon_k^{(s)}}}{\sum_{j=1}^K e^{\mu_{n,j} + s_n \epsilon_j^{(s)}}}, \tag{10}$$

where $\epsilon^{(s)} \sim \mathcal{N}(0, I)$, and to obtain the final expression we first reparametrize the integral to be an expectation under the standard normal and then apply Monte Carlo. To update $\gamma_n$ using SGD, we

---

[1] This is because the hyperparameter $\sigma^2$ in the approximate Gaussian likelihood is shared among all $K$ dimensions. If we choose a $\sigma_k^2$ per class the time and storage cost grows to $O(Km^2)$ which is too expensive.

[2] The hard constraint $\delta_n \geq 0$ is imposed through clipping.

---

**Algorithm 1** Kalman filter online learning

---

**Input:** Data stream $(x_n, y_n)_{n \geq 1}$; representation $\phi(x, \theta)$; parameters $\theta$; hyperparameters $(\sigma^2, \sigma_w^2)$
Initialise $M_0 = \mathbf{0}$, $A_0 = \sigma_w^2 I$ and $\delta_0$ (e.g. to value 0.0 so that $\gamma_0 = 1$)
**for** data point $n = 1, 2, 3, \ldots,$ **do**
    Observe $x_n$, compute $\phi_n = \phi(x_n, \theta)$ make a prediction for the label $y_n^*$ via equation 13
    Observe true class label $y_n$
    (Optional) Fine-tune $\theta$ together with the calibration parameter $\alpha$, as described in Appendix C
    Update $\delta_n$ as described in Section 2.1.1 and set $\gamma_n = e^{-0.5\delta_n}$
    $M_n^- = \gamma_n M_{n-1}$, $A_n^- = \gamma_n^2 A_{n-1} + (1 - \gamma_n^2)\sigma_w^2 I$
    Update Kalman statistics: $M_n = M_n^- + \frac{A_n^- \phi_n}{\sigma^2 + \phi_n^\top A_n^- \phi_n} \times (y_n^\top - \phi_n^\top M_n^-)$, $A_n = A_n^- - \frac{A_n^- \phi_n \phi_n^\top A_n^-}{\sigma^2 + \phi_n^\top A_n^- \phi_n}$
**end for**

---

differentiate this final Monte Carlo estimate of the loss $-\log p(y_{n,k} = 1 | y_{1:n-1})$ wrt the parameter $\delta_n$ in $\gamma_n = e^{-0.5\delta_n}$, where $\gamma_n$ appears in $(\mu_n, s_n^2)$ through the computation $M_n^- = \gamma_n M_{n-1}$, $A_n^- = \gamma_n^2 A_{n-1} + \sigma_w^2 (1 - \gamma_n^2)I$, while the parameters $(M_{n-1}, A_{n-1})$ of the posterior up to time $n - 1$ are taken as constants. Monte Carlo (MC) approximation can run efficiently by pre-computing $\mu_{n,k}$ and $s_n$ which overall takes $O(Km + m^2)$ time. Since these are scalar values, MC approximation complexity takes only $O(KS)$ time. Moreover, we found that significantly increasing $S$ did not make much difference in practice in terms of the variance of this approximation and we found that the moderate value $S = 50$ worked well in practice for the settings we considered.

## 3 RELATED WORK

**Online learning with Bayesian forgetting.** An alternative framework for non-stationary online learning can be based on the concept of Bayesian forgetting (BF) (Kulhavý and Zarrop, 1993; Honkela and Valpola, 2003; Graepel et al., 2010), as employed recently by Moens (2018) and Kurle et al. (2020); see also Li et al. (2021) for a related tempering based approach. The main idea is to allow some forgetting of the current posterior $p(w|y_{1:n})$ by creating a weighted geometric average with the prior, i.e. as $p_{\text{BF}}(w|y_{1:n}) \propto p(w|y_{1:n})^{\gamma_n} p_0(w)^{1-\gamma_n}$ (and re-normalize to get a new posterior for the next time step) where $\gamma_n \in [0, 1]$ is the forgetting factor. Most methods use non-learnable exponentially decaying values for the sequence $\gamma_n$, while Moens (2018) considers online variational inference. The BF approach differs significantly from the proposed Kalman filter (KF) algorithm where forgetting is achieved through convolving with the transition dynamics $p(w^-|y_{1:n}) = \int p(w^-|w)p(w|y_{1:n})dw$. In Appendix D, we derive a new algorithm following the BF principle that learns $\gamma_n$ online with SGD and we include this BF method in our experiments as an additional baseline compared to KF. An important computational difference between BF and KF is that, as shown in Appendix D, BF has cubic cost $O(m^3)$ per iteration as opposed to low quadratic $O(m^2)$ cost of the Kalman filter method.

**Related continual learning methods.** Classical approaches for continual learning have focused on reducing forgetting (De Lange et al., 2021; Parisi et al., 2019; Mai et al., 2022). More recently, the trend moved the focus towards forward transfer and efficient adaptation (Hadsell et al., 2020; Bornschein et al., 2022a; Ghunaim et al., 2023; Hayes and Kanan, 2022). While a significant progress has been achieved through this line of research, one major drawback, highlighted by different recent works (Ghunaim et al., 2023; Cai et al., 2021; Caccia et al., 2022), is the focus on small-scale artificial benchmarks with abrupt and unnatural distribution shift, and on metrics that fail to capture the capability of the models to efficiently adapt to the non-stationarity in the input data. In this work, we focus on fast adaptation and the next-step prediction problem from OL/OCL.

Different recent works considered the problem of fast adaptation and the set of constraints and objectives that are realistic in an online scenario. Different recent benchmarks, e.g. CLOC (Cai et al., 2021) and CLEAR (Lin et al., 2021), propose a sequence of temporally sorted images with naturally shifting visual concepts. CLOC leverages a subset of 39M images from YFCC100M (Thomee et al., 2016) along with their timestamps and geolocalisation tags, spanning a period of 9 years. The paper also highlights the limits of the classical continual learning approaches, and proposes simple baselines to overcome them, based mostly on adapting the online learning rate and the replay buffer size. They focus on an online evaluation protocol where each mini-batch is used for testing before adding it to the training dataset. In this work, we largely follow the CLOC setting. We evaluate our approach on the CLOC data and on modified CIFAR100 versions that are inspired by it.

Bornschein et al. (2022b) highlight the theoretical and practical benefits of using the online next-step performance for model evaluation. They emphasize the connection to compression based inference and to the prequential Minimum Description Length principle (Dawid and Vovk, 1999; Poland and Hutter, 2005). Empirically, they evaluate SGD based techniques on the pareto-front of prediction loss vs. computational requirements; and propose forward-calibration and specific rehearsal approaches to improve results. We compare against their methods in our experiments on CLOC.

The authors of Ghunaim et al. (2023) base their work on the observation that CL approaches have been developed under unrealistic constraints, allowing for offline learning on data with multiple passes without any limitation of computational cost. In realistic settings where we aim at updating a model on continuously incoming data, approaches that are slower than the stream would be impractical, or reach suboptimal performance in the limited time and computational budget. The authors propose an evaluation protocol that puts the computational cost at the center. They propose adding a delay between model update and evaluation. This delay is related to the computational cost of the updating method. Under the same budget, a twice more expensive approach would update the model half as often. With this realistic evaluation, the authors show that a simple baseline based on experience replay outperforms state-of-the-art CL methods, due to their complexity. In this work, we follow the protocol described in Ghunaim et al. (2023). Our approach has a relative complexity that requires no delay between training and evaluation. We therefore compare against the baselines reported in Ghunaim et al. (2023) that have the same properties, namely, ER (Chaudhry et al., 2019), ER++ (Ghunaim et al., 2023) and ACE (Caccia et al., 2022).

Finally, our method of learning $\gamma$ online could be seen as a way to perform change-point detection. However, unlike usual change-point detection methods (Page, 1957; Hawkins et al., 2003; Adams and MacKay, 2007; Fearnhead, 2006), learning $\gamma$ does not perform the hard change of the posterior parameter statistics, but rather operates in a soft way, gradually making the parameters to reset.

**Related Kalman filter methods.** The idea of relying on Kalman filters to estimate the posterior distribution over the weights of a neural networks dates back to the 90s, with work such as (e.g Singhal and Wu, 1988; Feldkamp et al., 1998; Puskorius and Feldkamp, 1994). The focus in these works is to accelerate learning by incorporating second order information in the step size, and the setting typically considered is that of stationary learning. Kalman filter for dealing with online learning has been explored in the linear case for example in Tsiamis and Pappas (2020); Kozdoba et al. (2019). In contrast, in our work we focus on the ability of Kalman filter to help with the online continual learning problem by relying on pretrained representations. In the concurrent work Chang et al. (2023), the authors proposed to use extended Kalman filter (EKF) directly on top of the neural network parameters assuming a similar drift in parameters as we did. One main difference with this work is that we only apply Kalman filter on the last layer and we assume that the covariance matrix is shared across all classes for the classification problems. This makes our method more scalable since it scales linearly with the number of classes and is only quadratic with respect to the dimensionality of the last layer. The method in Chang et al. (2023) scales as $O(P(L+K)^2)$ to update statistics, where $P$ is the number of parameters in the neural network, $L$ is at the order of 10 and $K$ is the number of classes, and requires $O(PL^2 + L^3)$ memory, which makes it a less scalable variant. Moreover, unlike Chang et al. (2023), in our work we also learn the forgetting parameter online.

## 4 EXPERIMENTS

### 4.1 ILLUSTRATIVE TIME SERIES EXAMPLE

We first apply our method to artificial time series data. The task is to track a non-stationary data stream of scalar noisy observations $y_n$ without any conditioning input $x_n$. We further assume a very simple model where the feature is just an univariate constant value equal to unity, i.e. $\phi_n = 1$ so that the observation likelihood simplifies as $p(y_n|w_n) = \mathcal{N}(y_n|w_n, \sigma^2)$ and the parameter $w_n$, to be inferred through time, models the unknown expected value of $y_n$.

Figure 1 shows the results of the Kalman model that was initialized with $\gamma_0 = 1$ and learns it online, as described in Appendix B. The non-stationary nature of the this series is such that the signal is piece-wise (noisy) constant with seven change-points. As shown by the second row in the left panel in Figure 1 the learned value of $\gamma_n$ is able to adjust to this non-stationarity by dropping the value of $\gamma_n$ quite below the value one (in order to refresh the Bayesian statistics over $w_n$) any time there is a change-point. Occasionally, $\gamma_n$ drops outside the task boundaries which could be explained by the outliers in the data. The drops of $\gamma_n$ at the task boundaries, however, are more durable giving the

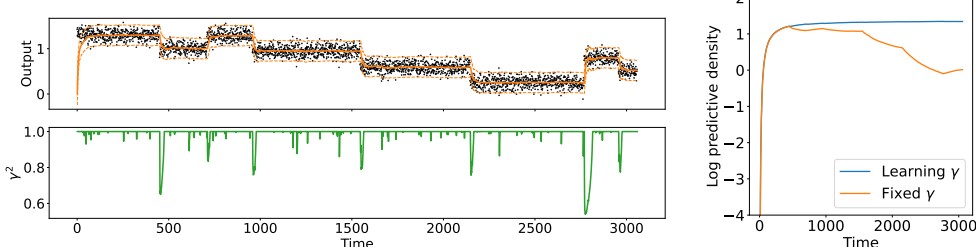

**Figure 1: Artificial time series example of** 3058 **observations.** Top row in left panel shows the data (black dots) and the predicted mean and uncertainty (orange lines) over $y_n$ (as data arrive sequentially from left to right and we perform online next step prediction), while the bottom row shows the optimized values of $\gamma_n^2 = e^{-\delta_n}$. Right panel shows the accumulated average log predictive density, i.e. $\frac{1}{n}\sum_{i=1}^{n} \log p(y_i|y_{1:i-1})$, computed across time for the model that learns $\gamma_n$ and the model that ignores non-stationarity by setting $\gamma_n = 1$ for all $n$.

posterior time to re-adjust to the distribution changes. In contrast, if we remove the ability to capture non-stationarity, i.e. by setting $\gamma_n = 1$ for all $n$, the performance gets much worse as shown by the accumulated log predictive density scores in Figure 1 and by Figure 4 in the Appendix E.

### 4.2 ONLINE CLASSIFICATION

In this section, we apply our method to online classification and we consider the OCL scenario where data arrive in small batches (chunks), see Cai et al. (2021); Bornschein et al. (2022b); Ghunaim et al. (2023): The learning algorithm is exposed to a data stream $\mathcal{S}$ such that every time step $n$, a chunk of data $S_n = \{(x_{n,b+i}, y_{n,b+i})\}_{i=1}^{b}$ of size $b$ is revealed. The algorithm first predicts the labels $y_s$, $s = (nb+1), \ldots, (n+1)b$, and then updates its parameters (see Algorithm 1) based on this chunk of data. As metric, we use *Average Online Accuracy*, $acc_o(n) = \frac{1}{nb}\sum_{s=1}^{nb} acc(y_s, \hat{y}_s)$, where $y_s$ is the ground truth label and $\hat{y}_s$ is the prediction of the model. This metric contains accuracies computed on the fly over training and quantifies how well the algorithm ingests new knowledge.

**Table 1: CIFAR-100** results. The numbers in **bold** correspond to the best performing method in the group.

| | Average Online Accuracy | |
| --- | --- | --- |
| Method | Stationary CIFAR-100 | Non-stationary CIFAR-100 |
| ***No backbone finetuning (Purely linear model)*** | | |
| Stationary Kalman Filter ($\gamma = 1.0$) | $11.63\% \pm 0.2\%$ | $13.73 \pm 0.9\%$ |
| Non-stationary Kalman Filter (fixed $\gamma = 0.999$) | $9.1\% \pm 0.1\%$ | $31.9\% \pm 0.8\%$ |
| Non-stationary Kalman Filter (learned $\gamma$) | $\mathbf{11.24\% \pm 0.1\%}$ | $\mathbf{33.3 \pm 0.3\%}$ |
| ***Backbone finetuning*** | | |
| Stationary Kalman Filter ($\gamma = 1.0$) | $17.24 \pm 0.2\%$ | $47.37\% \pm 0.9\%$ |
| Non-stationary Kalman Filter (fixed $\gamma = 0.999$) | $16.17\% \pm 0.2\%$ | $53\% \pm 0.9\%$ |
| Non-stationary Kalman Filter (learned $\gamma$) | $\mathbf{17.28\% \pm 0.2\%}$ | $\mathbf{53.47\% \pm 1\%}$ |
| ***Backbone finetuning with Replay*** | | |
| Stationary Kalman Filter ($\gamma = 1.0$) | $19.45\% \pm 0.4\%$ | $52.6\% \pm 1\%$ |
| Non-stationary Kalman Filter (fixed $\gamma = 0.999$) | $18.75\% \pm 0.1\%$ | $56.53\% \pm 1\%$ |
| Non-stationary Kalman Filter (learned $\gamma$) | $\mathbf{19.82\% \pm 0.3\%}$ | $\mathbf{57.13\% \pm 0.7\%}$ |
| **Baselines** | | |
| Online SGD | $16.25\% \pm 0.1\%$ | $49.5\% \pm 0.7\%$ |
| Online SGD + Replay | $16.88\% \pm 0.1\%$ | $54.5\% \pm 1\%$ |
| ER++ (Ghunaim et al., 2023) | $\mathbf{18.45\%}$ | $-$ |

### 4.2.1 ONLINE CLASSIFICATION ON CIFAR-100

We evaluate the performance of the Kalman Filter on two variants of online classification on CIFAR-100 (Krizhevsky and Hinton, 2009): *stationary online classifcation on CIFAR-100* and *non-stationary online classifaction on CIFAR-100*. In the *stationary case*, we follow the protocol described in Ghunaim et al. (2023) where the stream $\mathcal{S}$ is constructed by randomly shuffling CIFAR-100 dataset and split into chunks, so that the learning algorithm does one pass through CIFAR-100. This is

similar to One-Pass ImageNet (Hu et al., 2021) benchmark. Since it is randomly shuffled, there is no non-stationarity. In the *non-stationary case*, we consider a task-agnostic class-incremental version of Split-CIFAR100 as in Lee et al. (2020), where CIFAR-100 is split into 10 tasks (task identity is not communicated to the learning algorithm) each containing 10 different classes, concatenated into a stream and split into chunks. At any time, the learning algorithm solves a multi-classification problem with 100 classes. In this setting, there is very distinct non-stationarity related to the task changes. This benchmark is also studied by many class incremental continual learning approaches which focus on alleviating catastrophic forgetting (French, 1999; McCloskey and Cohen, 1989) and using average *incremental* accuracy as metric. Dealing with catastrophic forgetting goes beyond the scope of this work and we only focus on average *online* accuracy. In both cases, we follow training/evaluation protocol described in Ghunaim et al. (2023) dealing with chunks of size 10, using ResNet-18 (He et al., 2015) as feature extractor which is randomly initialized. For more details, see Appendix F.

The Kalman filter variants we consider are the following: *Stationary Kalman Filter ($\gamma = 1$)*, which could be seen as a form of Bayesian Logistic Regression, *Non-stationary Kalman Filter with fixed $\gamma = 0.999$* and *Non-stationary Kalman Filter with learned $\gamma$*. On top of that, we study performance of Kalman filter in three regimes: *no backbone finetuning*, a regime with fixed randomly initialized features $\phi$ which is a linear model, *backbone finetuning*, a regime where features $\phi$ are learned online and *backbone finetuning with replay*, a regime where on top of learning features $\phi$, we also replay previously data. As first baseline we consider online SGD with and without replay, which consists in sequential SGD algorithm. Also, for the stationary CIFAR-100, we consider the external baseline ER++ from Ghunaim et al. (2023), which is a version of Online SGD + Replay which does more update steps on the replay memory. This was the method with the highest performance as reported by Ghunaim et al. (2023). The hyperparameters for methods are selected by choosing the variant with highest cumulative log probabilities following MDL principle from Bornschein et al. (2022b). Replay setting is similar to the one considered in Ghunaim et al. (2023). See Appendix F for more details.

The results are given in Table 1. In the **stationary** CIFAR-100 case, we observe that stationary Kalman filter provides a reasonable performance and is generally better than non-stationary Kalman filter with fixed $\gamma$. This is consistent with our intuition that there is not much non-stationarity to model and therefore we would not expect Kalman filter to help. We see that Kalman Filter with learning $\gamma$ leads to slightly better results than stationary case. Intuitively, it makes sense since in the worst case, this variant could revert back to $\gamma = 1$. Moreover, replay-free Kalman filter leads to very competitive results against external baselines. Adding replay to Kalman improves results even further, beating ER++ baseline which uses much more replay than ER; see Ghunaim et al. (2023). Kalman filter performs consistently better than Online SGD in this setting. In the **non-stationary** CIFAR-100

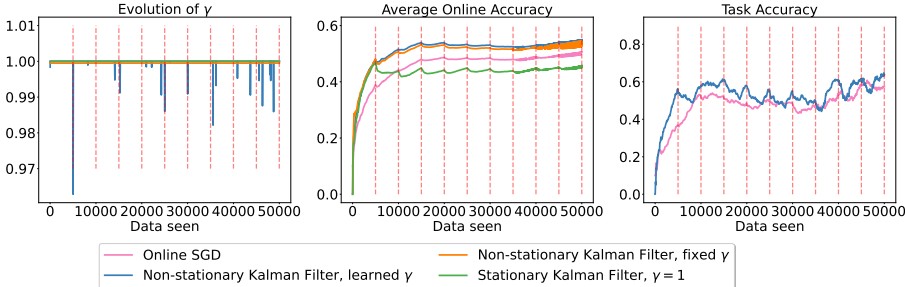

**Figure 2: Non-stationary CIFAR-100**. Left - evolution of $\gamma$, center - average online accuracy, right - smoothed task accuracy (5% of data). Dashed lines indicate task boundaries.

case, stationary Kalman filter performs consistently worse than its non-stationary variants. This is consistent with our intuition since in this case, there is non-stationarity which Kalman filter can capture. Moreover, again learning $\gamma$ generally leads to better performance. Figure 2, left, visualizes the dynamics of learning $\gamma$ for the *Backbone Finetuning* setting, with red dashed lines indicating task boundaries. In many cases, $\gamma$ drops at task boundaries, pushing down probabilities of classes from previous classes and focusing more on future data. This is the desired behaviour of the method since it essentially captures non-stationarity. Furthermore, in some cases $\gamma$ could drop slightly outside of task boundaries which could be explained by the presence of outliers. This, however, does not make the performance of the method worse. On top of that, Online SGD also achieves strong performance despite being a simple method. Kalman filter, however, achieves superior performance to Online

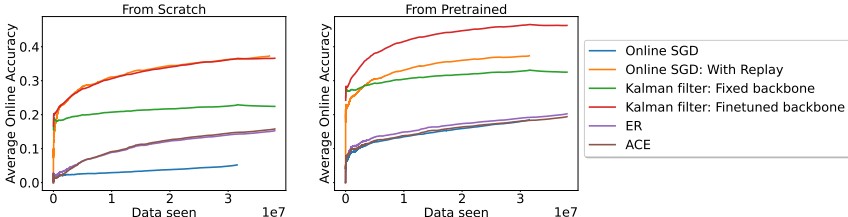

**Figure 3: CLOC** results. Left - learning from scratch, right - learning from pretrained model. External baselines are taken from Ghunaim et al. (2023): ER and ACE (Caccia et al., 2022). On the left, Online SGD with replay and Kalman filter (KF) with finetuned backbone are overlapping, while KF performs much better on the right.

SGD. We see that replay improves performance for both methods and Kalman filter performance can be further improved if replay is allowed. In Figure 2, right, we compare performance of the Kalman filter (learned $\gamma$ and finetuned backbone) against Online SGD. We see that overall Kalman filter achieves high accuracy on each task faster than Online SGD making this method more data efficient.

### 4.2.2 ONLINE LARGE-SCALE CLASSIFICATION ON CLOC

In CLOC (Cai et al., 2021), each image in a chronological data-sequence is associated with the geographical location where it was taken, discretized to 713 (balanced) classes. It is a highly non-stationary task on multiple overlapping time-scales because, e.g., major sports events lead to busts of photos from certain locations; seasonal changes effect the appearance of landmarks; locations become more or less popular over time; etc. We use the version of CLOC described in Bornschein et al. (2022b): around 5% of the images could not be downloaded or decoded which leaves us with a sequence of 37,093,769 images. This version of the dataset is similar to the one considered in Ghunaim et al. (2023), but we are mindful of potential small differences due to the downloading errors. We follow the same protocol as in Ghunaim et al. (2023) and in Bornschein et al. (2022b).

We use a ResNet-50 backbone and receive the data in chunks of 128 examples. For the Kalman filter, we consider the variant with learned $\gamma$, which performed always better than any fixed one, including $\gamma = 1$. We either keep the backbone fixed, or finetune it. The case of fixed backbone corresponds to a linear model only. For more details and hyperparameters selection, see Appendix F. We run experiments where we either start learning on CLOC from scratch, or start with a ImageNet-pretrained backbone via supervised loss. As baselines, we consider Online SGD with and without replay from Bornschein et al. (2022b), and compare to the results from Ghunaim et al. (2023): ER Chaudhry et al. (2019) and ACE (Caccia et al., 2022), which were the best performing baselines on this setting as reported by Ghunaim et al. (2023). To produce the plots, we asked the authors in (Ghunaim et al., 2023) to provide us the data from their experiments. The results are shown in Figure 3, where Kalman filter provides very strong performance compared to the baselines. When learning from scratch, replay-free Kalman filter matches the performance of Online SGD with replay. This is a strong result since Kalman filter does not need to store additional data in memory. Even having Kalman filter with fixed backbone performs much better than Online SGD. When starting from pretrained model, Kalman filter manages to learn more efficiently than any of the baselines. Moreover, finetuning the backbone provides a large boost in performance compared to using a fixed backbone. We see that in case of learning from scratch, using online SGD with Replay performs similarly to finetuning a backbone together with a Kalman filter. We believe this is due to the fact that the random backbone parameters are too noisy and when combined with Kalman filter updates, the whole process still remains too noisy. The method, however, still performs significantly better than online gradient descent without replay. For more details about backbone finetuning, see Appendix G.

## 5 CONCLUSION

We presented a probabilistic online learning method that combines efficient Kalman filter inference with online learning of deep learning representations. We have demonstrated that this method is able to adapt to non-stationarity of the data and it can give competitive next time-step data predictions. Some directions for future research are: Firstly, it would be useful to investigate other approximate inference methods for the classification case where we may construct a more accurate, but still computationally efficient, online Gaussian approximation to the softmax. Secondly, it would be interesting to extend the transition dynamics to include higher order Markov terms, which could increase the flexibility of the algorithm to model more complex forms of non-stationarity and distribution drift.

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

## A  CONNECTION WITH ONLINE BAYESIAN LINEAR REGRESSION

When the forgetting coefficient is $\gamma_n = 1$ for all $n$, the stochasticity in the transitions is removed since the parameter transition density becomes a point mass, i.e. $p(w_n|w_{n-1}) = \delta(w_n - w_{n-1})$. Then, the Kalman filter reduces to updating the Gaussian posterior density $p(w|y_{1:n}) = \mathcal{N}(w|m_n, A_n)$ where

$$m_n = m_{n-1} + \frac{A_{n-1}\phi_n}{\sigma^2 + \phi_n^\top A_{n-1}\phi_n}(y_n - \phi_n^\top m_{n-1}), \quad A_n = A_{n-1} - \frac{A_{n-1}\phi_n\phi_n^\top A_{n-1}}{\sigma^2 + \phi_n^\top A_{n-1}\phi_n}, \quad (11)$$

with the initial conditions $m_0 = 0$ and $A_0 = \sigma_w^2 I$. We see that these recursions compute exactly in an efficient $O(m^2)$ time the standard Bayesian linear regression posterior given by $p(w|y_{1:n}) = \mathcal{N}(w|A_n\sigma^{-2}\sum_{i=1}^n \phi_i y_i, A_n)$, with $A_n = \left(\sigma^{-2}\sum_{i=1}^n \phi_i\phi_i^\top + \sigma_w^{-2}I\right)^{-1}$, and where the connection with the online updates can be seen by applying the Woodbury matrix identity. Obtaining the online Bayesian linear regression recursion as a special of a Kalman filter is well known in the literature; see for example Section 3.2 in the book of Särkkä (2013).

## B  ONLINE UPDATING THE FORGETTING COEFFICIENT $\gamma_n$ FOR REGRESSION

Learning online $\gamma_n$ for the regression case is simpler than in classification since now the predictive density is a tractable Gaussian given by equation 5. Thus, given that we parametrize $\gamma_n = e^{-0.5\delta_n}$, with $\delta_n \geq 0$, the update for $\delta_n$ is written as

$$\delta_n \leftarrow \delta_n + \rho_n \nabla_{\delta_n} \log \mathcal{N}(y_n|\phi_n^\top m_n^-, \phi_n^\top A_n^- \phi_n + \sigma^2). \quad (12)$$

The dependence of the log density on $\delta_n$ is through the parameters $(m_n^-, A_n^-)$ given by equation 3 while the parameters $(m_{n-1}, A_{n-1})$ of the posterior up to time $n-1$ are taken as constants.

## C  CALIBRATION OF CLASS PROBABILITIES AND FINE-TUNING THE REPRESENTATION

Given that in classification the predictive posterior $q(W_n|y_{1:n-1})$ is an (possibly crude) approximation to the true posterior, the log predictive probability estimate in (12) can be inaccurate. We improve this estimate by fine-tuning over a *calibration parameter* that is optimized with gradient steps. In the same step we can also fine-tune the neural network parameters $\theta$ that determine the representation vector $\phi(x, \theta)$. Specifically, for the calibration procedure we introduce a parameter $\alpha > 0$ that rescales the logits inside the softmax function, so that the softmax in equation 10 is replaced by $\frac{e^{\alpha f_{n,k}}}{\sum_{j=1}^K e^{\alpha f_{n,j}}}$ and the final Monte Carlo estimate becomes

$$\log p(y_{n,k} = 1|y_{1:n-1}, \alpha, \theta) \approx \log \frac{1}{S}\sum_{s=1}^S \frac{e^{\alpha\mu_{n,k}+\alpha s_n\epsilon_k^{(s)}}}{\sum_{j=1}^K e^{\alpha\mu_{n,j}+\alpha s_n\epsilon_j^{(s)}}}. \quad (13)$$

Then the negative log predictive probability, $-\log p(y_{n,k} = 1|y_{1:n-1}, \alpha, \theta)$, is treated as a loss that is optimized jointly over $(\alpha, \theta)$ with online SGD steps, i.e. as individual data points $(x_n, y_n)$ arrive sequentially. This resembles the forward-calibration described in (Bornschein et al., 2022b). Algorithm 1 summarizes the whole online learning procedure that includes the Kalman filter recursion, update of coefficient $\gamma_n$ and fine-tuning of the representation parameters $\theta$ and calibration parameter $\alpha$. An ablation study in Appendix G shows that the calibration procedure can significantly improve the log predictive probability estimates.

The fully online process for fine-tuning the deep network parameters $\theta$ in Algorithm 1 can be expensive since modern hardware is computationally more effective when forward and backpropagation passes in deep neural nets are applied jointly to minibatches rather to individual data points. In practice, we therefore also consider a faster version of Algorithm 1 where we predict ahead a batch of $b$ data points by treating them as i.i.d. and then take a gradient step over $(\alpha, \theta)$, i.e. we use as loss $-\frac{1}{b}\sum_{i=1}^b \log p(y_{n+i}|y_{1:n-1}, \alpha, \theta), i = 1, \ldots, b$. This creates two options for the Markov dynamics: (i) either apply the transition "batch-wise" where the transition is taken every $b$ data forming the

minibatch (so that time index $n$ corresponds to the number of minibatches), or (ii) the minibatch updating only affects the prediction over future data (we predict ahead $b$ points instead of only the next one) and the fine-tuning of $(\alpha, \theta)$, while the subsequent Kalman recursion steps and SGD update over $\gamma_n$ are applied online, i.e. by processing the $b$ data in the minibatch one by one. Both schemes can be useful in practice, as further discussed in the Appendix. For simplicity, Algorithm 1 presents the purely online version, while pseudocode for the above minibatch-based variants is similar.

## D    ONLINE LEARNING WITH BAYESIAN FORGETTING

We develop an alternative non-stationary online method based on the concept of Bayesian forgetting (BF) (Kulhavý and Zarrop, 1993; Honkela and Valpola, 2003; Graepel et al., 2010; Moens, 2018; Kurle et al., 2020). We first derive this method and we contrast it with our Kalman filter approach described in the main text. To simplify notation we remove the index $n$ when referring to quantifies $w, \gamma, A, m$. In some part description next we shall use a more general prior of the form $p_0(w) = \mathcal{N}(w|\mu_w, \Sigma_w)$. While both BF and Kalman filter recursively estimate some Gaussian posterior distribution $p(w|y_{1:n}) = \mathcal{N}(w|m, A)$ as the data arrive by incorporating forgetting with a factor $\gamma \in [0, 1]$, they do this differently. Recall that in the Kalman filter method forgetting is applied through convolving with the transition/diffusion dynamics $p(w^-|w) = \mathcal{N}(w^-|\gamma w + (1-\gamma)\mu_w, (1-\gamma^2)\Sigma_w)$ so that

$$p(w^-|y_{1:n}) = \int p(w^-|w)p(w|y_{1:n})dw = \mathcal{N}(w'|\gamma m + (1-\gamma)\mu_w, \gamma^2 A + (1-\gamma^2)\Sigma_w),$$

where in the Kalman filter terminology this is referred to as the prediction step. In contrast, a corresponding step in BF involves mixing the current posterior with the prior using a weighted geometric average and then re-normalizating, as follows

$$p_{\text{BF}}(w^-|y_{1:n}) = \frac{p(w|y_{1:n})^\gamma p_0(w)^{1-\gamma}}{\int p(w'|y_{1:n})^\gamma p_0(w')^{1-\gamma}dw'} = \mathcal{N}(w|\mu', A'),$$

where

$$A' = \left(\gamma A^{-1} + (1-\gamma)\Sigma_w^{-1}\right)^{-1},$$

$$\mu' = \left(\gamma A^{-1} + (1-\gamma)\Sigma_w^{-1}\right)^{-1}\left(\gamma A^{-1}m + (1-\gamma)\Sigma_w^{-1}\mu_w\right).$$

The remaining steps for the BF method remain the same as in the Kalman filter method. For instance, the posterior update to incorporate the next data point $y_{n+1}$ involves the Bayes' rule $p(w|y_{1:n+1}) \propto p(y_{n+1}|w)p_{\text{BF}}(w|y_{1:n})$ where $p_{\text{BF}}(w|y_{1:n}) := p_{\text{BF}}(w^-|y_{1:n})$ while the online update for $\gamma$ is done by taking an SGD step to maximize the corresponding log predictive likelihood computed as in equation 10.

Despite that BF and Kalman filter differ only on how forgetting is incorporated, this still results in rather different properties. Firstly, the Kalman algorithm corresponds to a precise probabilistic state space model, where forgetting is achieved through the Markov transition dynamics. In contrast, in BF the forgetting is incorporated heuristically by creating a weighted geometric average between the posterior and the prior. Secondly, and more importantly in practice, the Kalman filter method scales as $O(m^2)$ per iteration but BF scales cubically as $O(m^3)$. To clarify this, firstly observe that an efficient way to update online $p_{\text{BF}}$ is to propagate a forward (including also the step for incorporating the next observation $y_{n+1}$) a recursion over the natural parameters in quadratic $O(m^2)$ time. These natural parameters are the inverse covariance, i.e. $\gamma A^{-1} + (1-\gamma)\Sigma_w^{-1}$, and the inverse covariance times the mean vector, i.e. $\gamma A^{-1}m + (1-\gamma)\Sigma_w^{-1}\mu_w$. To give the exact form of this recursion for the classification case let us simplify the prior as $\mathcal{N}(w|\mu_w, \Sigma_w) = \mathcal{N}(w|0, \sigma_w^2 I)$, i.e. consider the one used in the main text and all our experiments. Then the steps are the following:

1. Initialize the natural parameters as $\Lambda = \frac{1}{\sigma_w^2}I$ ($m \times m$ matrix) and $\lambda = \mathbf{0}$ ($m \times k$ matrix).

2. Take the weighted geometric average between posterior and prior and re-normalize. This gives a new Gaussian with natural parameters

$$\Lambda' = \gamma\Lambda + (1-\gamma)\frac{1}{\sigma_w^2}I, \quad \lambda' = \gamma\lambda.$$

3. Incorporate the next data point to obtain a posterior Gaussian with natural parameters

$$\Lambda = \Lambda' + \frac{1}{\sigma^2}\phi \times \phi^\top, \quad \lambda = \lambda' + \frac{1}{\sigma^2}\phi \times y^\top,$$

where recall that $y$ is a $k \times 1$ one hot vector and $\phi$ a $m \times 1$ vector and for simplicity we do not denote the data index.

However, the above efficient updates over the natural parameters cannot prevent cubic cost because we also need to predict the next data output and learn online the forgetting coefficient $\gamma$. For that, we need to compute the predictive distribution (as in equation 10 in the main text) which requires the actual (and not the inverse) covariance matrix $A^- = (\Lambda')^{-1}$ as well as the mean vector. Based on the above expressions this has $O(m^3)$ cost per iteration due to the inverse $A' = (\Lambda')^{-1} = \left(\gamma\Lambda + (1-\gamma)\sigma_w^{-2}I\right)^{-1}$. More precisely, we can see that the actual predictive posterior density over the $k$ logits is a factorised Gaussian with mean vector $\mu = (\lambda')^\top (\Lambda')^{-1}\phi$ and isotropic/shared variance $s^2 = \phi^\top (\Lambda')^{-1}\phi$. For these last two computations we need to perform an $O(m^3)$ Cholesky decomposition of the matrix $\Lambda'$.

The $O(m^3)$ cost of BF could be too high in some of the experiments where the size $m$ of the feature vector is typically of order of thousands, e.g. $m = 2048$ for the CLOC dataset.

We compare our method with BF on CLOC dataset. In Figure 5, we compare Bayesian Forgetting method against Kalman Filter, where we use a fixed pretrained backbone (similar to the main experiment on CLOC), but we see only 1 example at a time. On top of that, we run a large scale experiment on CLOC similar to the one presented in Figure 3 from the main paper. In this case, similarly, we see a chunk of 128 points at a time and we allow for using a fixed backbone or for learning one. The results are given in Figure 6. In both cases, we see that KF consistently outperforms Bayesian Forgetting whilst being also a much more efficient method.

## E  FURTHER DETAILS ABOUT THE TIME SERIES EXAMPLE

We generated the time series dataset consisted of 3058 observations sequentially by using 8 segments with mean values $\{1.3, 1.0, 1.3, 0.95, 0.6, 0.25, 0.8, 0.5\}$ and where the change-points between these segments occurred (randomly) at the following 7 time steps $\{451, 709, 958, 1547, 2147, 2769, 2957\}$. To obtain each observed $y_n$ we added Gaussian noise with variance $0.01$.

For these time series data we applied a modification of the Kalman updates so that all updates remain the same, except for the update of the parameter $m_n^-$ which now does not shrink to zero and it has the form

$$m_n^- = m_{n-1}, \quad \text{(while before was } m_n^- = \gamma_n m_{n-1}).$$

This is appropriate in this case because shrinking the predictive posterior mean to zero by multiplying it by $\gamma_n \le 1$ is a very strong prior assumption that $w_n$ has zero mean which does not hold, since the time series data can have arbitrary values away from zero. A more formal justification of the above is that the Markov dynamics have now the form $p(w_0) = \mathcal{N}(w_0|0, \sigma_w^2 I)$ and $p(w_n|w_{n-1}) = \mathcal{N}(w_n|(1-\gamma_n)\mu_{n-1} + \gamma_n w_{n-1}, \sigma_w^2(1-\gamma_n^2)I)$ and each mean parameter $\mu_{n-1}$ when we transit from time step $n-1$ to $n$ is found by empirical Bayes so that $\mu_{n-1} = m_{n-1}$ and where $m_{n-1}$ is the mean of $p(w_{n-1}|y_{1:n-1})$.

The hypermarameters in the experiment were set to the following values: $\sigma_w^2 = 0.01, \sigma^2 = 0.05$ while $\delta_0$ was intitialized to $0.0$ (so that $\gamma_0 = 1.0$) and then updated at each step by performing SGD steps with learning rate equal to $1.0$.

Figure 1 in the main paper shows the results when $\gamma_n$ is updated online. For comparison, in Figure 4 we show the online predictions for the case of having fixed $\gamma_n = 1$ for any $n$, so that the ability to model non-stationarity is removed. Clearly, when not learning $\gamma_n$ the model is not able to adjust to non-stationarity.

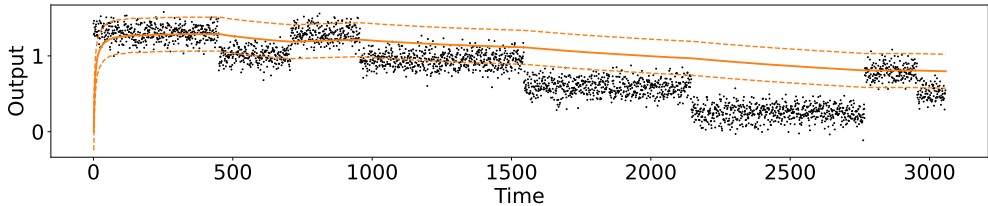

**Figure 4:** Online prediction on the artificial time series example by applying the Kalman filter model with fixed $\gamma_n = 1$.

# F    EXPERIMENTAL DETAILS AND HYPERPARAMETERS SELECTION

## F.1    HYPERPARAMETERS SELECTION

For hyperparameters we do the grid search over ranges. The hyperparameters for methods are selected by choosing the variant with highest cumulative log probabilities following MDL principle from (Bornschein et al., 2022b). When finetuning backbone together with learning calibration parameter $\alpha$, we considered a different relative scaling $\beta$ of the gradients of the backbone compared to $\alpha$ - meaning that if we use learning rate $\eta$ for $\alpha$, the learning rate for backbone is $\beta * \eta$. In case where we learned $\gamma$, we initialized $\gamma$ at $\gamma_{init}$ which is also a hyper-parameter. For each of the method, we optimized the hyper-parameters separately. The hyper-parameters and the considered values are given in the Table 2.

**Table 2:** Hyper-parameters and considered values.

| Hyperparameter | Considered values |
| --- | --- |
| Learning rate for backbone and $\alpha$, $\eta$ | [0.1, 0.01, 0.001, 0.0005, 0.0001] |
| Learning rate for $\gamma$, $\eta_\gamma$ | [0.5, 0.1, 0.01, 0.001] |
| Relative backbone-$\alpha$ gradient scaling, $\beta$ | [0.01, 0.1, 1., 10., 100.0] |
| Initial $\gamma_{init}$ | [0.9, 0.99, 0.999, 1.0] |

## F.2    FINETUNING PROTOCOL

The algorithm 1 describes Kalman filter online learning where the learning happens for each data point. We adapt this algorithm for chunk-based learning in the following way. When we receive the chunk of data of certain size, we use the available Kalman statistics in order to calculate predictive log-probabilities (see Appendix C) for each data point in the chunk **independently**. After that, we do the gradient update on the backbone and $\alpha$, leading us new backbone parameters and new $\alpha$. After that, we re-compute the features on the same chunk of data and we do Kalman recursion on this chunk going through it sequentially. This latter process only involves updating statistics and $\gamma_n$ and doesn't involve backbone or $\alpha$ finetuning. When we do this Kalman recursion, we consider two options on how to apply **Markov transition**. In the first option, **Always Markov**, we apply Markov transition for every data point during Kalman recursion. In the second option, **Last Step Markov**, we only do Markov transition on the last point in the chunk. The difference basically lies in what we consider a non-stationary data point. In case of **Always Markov**, it is each data point in the chunk. In case of **Last Step Markov**, it is the whole chunk. We found that for different scenarios, different strategies led to different results.

## F.3    ADDITIONAL PER-BENCHMARK PARAMETERS

On top of the hyper-parameters which we selected for each of the baseline separately, we also found that there was a group of hyper-parameters which generally provided consistently good results for all the baselines for each benchmark. These parameters are the following:

- Transition type: **Always Markov** or **Last Step Markov**

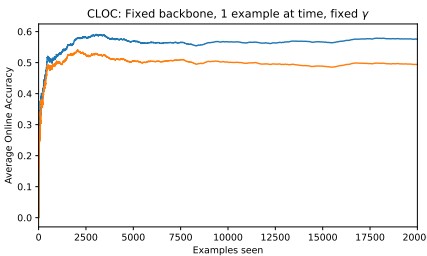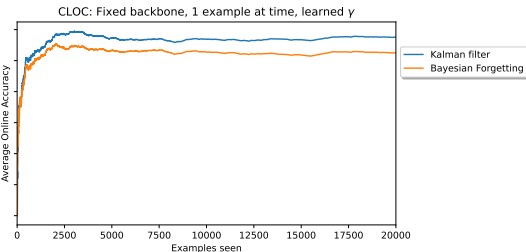

**Figure 5:** CLOC results in comparing BF with KF methods. In this experiment the chunk size is 1, and we use a fixed pre-trained backbone for both methods.

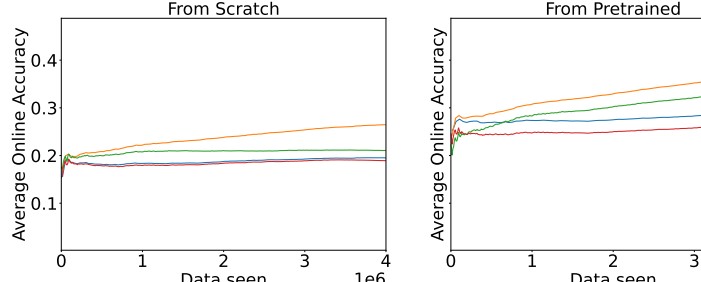

**Figure 6:** CLOC results in comparing BF with KF methods. In this experiment the chunk size is 1, and we use a fixed pre-trained backbone for both methods.

- Bias in features: Whether we add bias to the features or not
- Features normalization: Whether we normalize features vector, i.e., whether we divide it by the square root of its dimensionality.
- Optimization algorithm: plain SGD or Adam with weight decay ($\lambda$)

These parameters were found empirically and similarly chosen via MDL principle. The table 3 gives the summary.

**Table 3:** Per benchmark parameters.

| Benchmark | Transition type | Use Bias | Use normalization | Optimization alg. |
|---|---|---|---|---|
| Stationary CIFAR-100 | Last Step Markov | No | Yes | SGD |
| Non-Stationary CIFAR-100 | Always Markov | Yes | Yes | SGD |
| CLOC with pretrained backbone | Always Markov | No | No | SGD |
| CLOC from scratch | Last Step Markov | No | No | AdamW ($\lambda = 10^{-4}$) |

## F.4 CIFAR-100 EXPERIMENTS

In both cases of CIFAR-100 experiments, we follow protocol described in Ghunaim et al. (2023). We use ResNet-18 as backbone model and SGD as learning algorithm. We split data into chunks of size 10. In some cases, we allow a replay strategy suggested in Ghunaim et al. (2023), where we keep a memory of size 100 containing most recently seen examples. Then, for each learning iteration, we sample a chunk of size 10 from memory and append it to the right of the current chunk, creating a chunk of size 20.

## F.5 CLOC EXPERIMENTS

Similar to Ghunaim et al. (2023) and Bornschein et al. (2022b), we considered ResNet-50 as backbone and chunk size equal to 128. Empirically, we found that SGD optimization worked much better

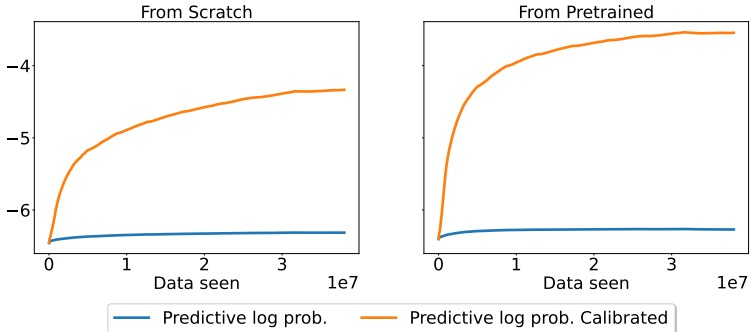

**Figure 7: CLOC** log probabilitites for Kalman filter with finetuned backbone and finetuned delta. We show the data (black dots) and the predicted mean and uncertainty (orange lines) over $y_n$ (as data arrive sequentially from left to right and we perform online next time step prediction).

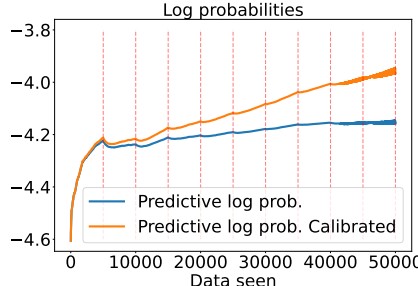

**Figure 8: Non-stationary CIFAR-100** log probabilitites for Kalman filter with finetuned backbone and finetuned $\delta$.

for pre-trained model whereas Adam with weight decay ($\lambda = 10^{-4}$) was working much better for learning from scratch. Moreover, we also found that when learning from scratch, it worked much better if we used Last Step Markov transition (see Appendix F.2). For pre-trained model, it worked better use Always Markov transition (see Appendix F.2).

For baselines, we considered Online SGD baseline from Bornschein et al. (2022b), with EMA parameter equal to 1 for fair comparison. The case of online SGD with replay corresponds to using 8 replay streams on top of the learning stream which make sure that the data distribution in the replay buffer is well behaved (see Bornschein et al. (2022b) for more details).

## G   THE EFFECT OF CALIBRATION ON ONLINE CLASSIFICATION

In Appendix C, we describe a procedure to finetune the model backbone $\phi$ as well as to finetune the parameter $\alpha$ which affects the predictive log probability. This parameter $\alpha$ essentially allows us to calibrate the predictive log probabilities. The effect of this calibration is shown in Figure 7 for CLOC dataset and in Figure 8 for non-stationary CIFAR-100. In both cases, we use the version of Kalman filter which finetunes the backbone and learns $\gamma$. We see a very drastic positive effect of calibration – calibrated log probabilities become much higher. Related to the discussion of prequential MDL (Bornschein et al., 2022b), it essentially allows to have a model with lower description length. Moreover, since we use predictive log probability as a learning signal for backbone, the calibration allows to tune the scaling of this term.

## H   CHUNK SIZE ABLATION

In this section we provide the ablation of the impact of the chunk size on the average online accuracy. We provide it for CIFAR-100 as well as for CLOC. We use the Kalman filter variant with learned $\gamma$ with either fixed or finetuned backbone. The results are given in Figure 9.

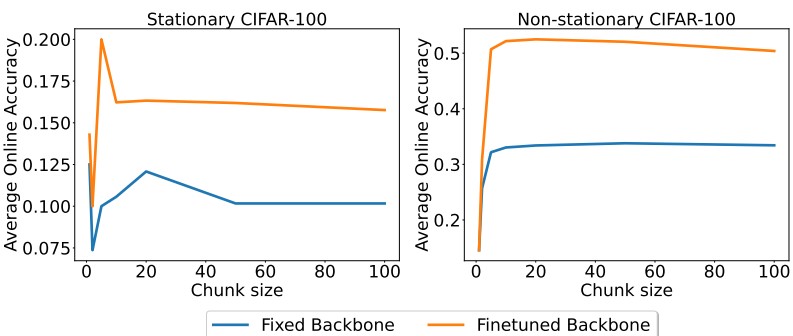

**Figure 9:** Chunk size ablation for CIFAR-100.

