# OpenReview forum: "Kalman Filter for Online Classification of Non-Stationary Data"
_ICLR.cc/2024/Conference — ICLR 2024 poster_

### Official Review · Reviewer_VM8e · 2023-10-30

**Soundness:** 3 good
**Presentation:** 3 good
**Contribution:** 2 fair
**Rating:** 8
**Confidence:** 3

**Summary:**

The paper studies the challenging problem of online continual learning and introduces an approach based on the Kalman filter (KF).  This paper models non-stationary as parameter drift based on linear stochastic dynamics and employs KF to estimate the posterior distribution of adaptable model parameters. Additionally, the paper discusses updates to the forgetting coefficient of KF. The experimental results demonstrate the superior performance of the proposed method compared to previous online continual learning methods.

**Strengths:**

The paper studies a practical and interesting problem: online continual learning, emphasizing non-stationary. It explores the application of  KF in classification problems by updating the forgetting coefficient.

The proposed method, which utilizes a Kalman filter-based approach for prior and posterior estimation of model parameters, presents a reasonable solution to tackle the (non-stationary)  online learning problem.

Overall, the paper is well-structured and effectively communicates the details of the proposed method.

Furthermore, in the experimental evaluation, the proposed method demonstrates superior performance compared to previous approaches, as reported in the paper.

**Weaknesses:**

1) This paper could benefit from a comparison with related works that utilize the Kalman filter for online continual learning and adaptation tasks, as seen in [1]. Such a comparison or discussion regarding the difference between the proposed method and [1] would enhance the paper's comprehensiveness.

2) The paper assumes that the ground-truth model follows a linear predictor based on features,  followed by adapting the last layer of the neural network using KF. The results in Table 1 suggest that the "Backbone finetuning" method outperforms "No backbone finetuning (Purely linear model)." This observation raises questions about the validity of the assumption regarding linear stochastic dynamics.  It is essential to engage in an in-depth discussion regarding the correctness and practicality of these assumptions.

3) In the evaluation of CLOC in section 4.2.2, it's crucial to consider the real-time computational cost for each algorithm, as emphasized in [2]: In the evaluation of the proposed method could be a prudent step.

[1] A. Abuduweili, et al, “Robust online model adaptation by extended kalman filter with exponential moving average and dynamic multi-epoch strategy”, L4DC 2020.
[2] Bornschein, et al. “Sequential learning of neural networks for prequential mdl.” ICLR, 2022.

**Questions:**

1) How does the proposed method compare to related Kalman filter-based techniques for online continual learning and adaptation, such as KF in  [1]?

2) Does the assumption concerning linear stochastic dynamics align with the outcomes presented in Table 1, given that the "Backbone finetuning" method outperforms the "No backbone finetuning (Purely linear model)"?

3) What are the real-time computational costs associated with each algorithm?

[1] A. Abuduweili, et al, “Robust online model adaptation by extended kalman filter with exponential moving average and dynamic multi-epoch strategy”, L4DC 2020.

---

> ### Author Response · Authors · 2023-11-22
> **Response to VM8e: part 1**
>
> We would like to thank reviewer VM8e for the feedback. Please find our detailed answer below.
>
> > This paper could benefit from a comparison with related works that utilize the Kalman filter for online continual learning and adaptation tasks, as seen in [1]. Such a comparison or discussion regarding the difference between the proposed method and [1] would enhance the paper's comprehensiveness. How does the proposed method compare to related Kalman filter-based techniques for online continual learning and adaptation, such as KF in [1]?
>
> We would like to thank the reviewer for suggesting this paper. Another relevant paper is also based on Extended Kalman Filter (EKF) from Chang et al., which we discuss in our paper. The drawback of the paper you proposed and this other EKF work, consists in fact that these approaches scale poorly with problem dimensionality. EKF from Chang et al. has complexity $O(PC^2)$, where P is the number of parameters and C is the number of classes, so it is not  applicable in situations where C is large such as CLOC. Note that our method has complexity $O(mK+m^2)$ on top of online SGD complexity (or $O(PC+P^2)$  using Chang et al notation) so in practice runs as fast as online SGD which has $O(P K)$ cost.  Further, the paper you mentioned in [1], uses an ordinary EKF approach with additional modifications, which scales as $O(K^3+PK^2+P^2K)$. It is easy to see from the algorithm 2, where $H_t$ is the matrix of size $K \times P$ and $P_t$ is the matrix of size $P \times P$. Line 2 requires $H_t*P_{t-1}*H_t^T$ multiplication which is done in $O(KP^2+K^2 P)$. After that we need to invert the matrix of size $K \times K$ which takes $O(K^3)$. Therefore the total running complexity is $O(K^3+PK^2+P^2K)$. For large $K$ the term is dominated by $O(K^3+PK^2)$ and for large $P$ it is dominated by $O(P^2K)$. Moreover, the method requires us to store the matrix $P$ as well as to store (during computations) $H_t$, this results in $O(P^2+KP)$ storage complexity. Note that in Neural Networks settings, P is of the order of millions / billions. Our algorithm requires us to store $O(m^2+Km)$ which corresponds to the last layer covariance matrix and per-class mean vectors. Therefore, our algorithm is significantly cheaper than this variant, both in terms of computation time and memory. Even if we were to apply this approach to the last layer, we would still get a very large complexity of $O(K^3+mK^2+m^2 K)$, which would make this approach less practical than ours. Based on this thinking, comparing to the proposed approach might not be very informative, since we are also primarily interested in scalability.
>
> > The paper assumes that the ground-truth model follows a linear predictor based on features, followed by adapting the last layer of the neural network using KF. The results in Table 1 suggest that the "Backbone finetuning" method outperforms "No backbone finetuning (Purely linear model)." This observation raises questions about the validity of the assumption regarding linear stochastic dynamics. It is essential to engage in an in-depth discussion regarding the correctness and practicality of these assumptions. Does the assumption concerning linear stochastic dynamics align with the outcomes presented in Table 1, given that the "Backbone finetuning" method outperforms the "No backbone finetuning (Purely linear model)"?
>
> We formulate the method assuming this linear model, but later on, we indeed, violate this assumption. Nevertheless, we still use the KF updates assuming the linear model and we see that in practice this works very well. That said, what we are doing is not completely unreasonable. If the model is fully linear, then the KF updates are EXACT, meaning that we have true posterior over the last layer at time t. If the model parameters change (thetas or gammas), the updates stop being exact, but become approximate and we in fact propagate the approximate posterior. Such a setting is well studied and is known as online Bayesian learning. Without explicitly saying that, this is in fact what is happening in our case when the model parameters change. We will add an explicit comment about it in the text.

---

> ### Author Response · Authors · 2023-11-22
> **Response to VM8e: part 2**
>
> > In the evaluation of CLOC in section 4.2.2, it's crucial to consider the real-time computational cost for each algorithm, as emphasized in [2]: In the evaluation of the proposed method could be a prudent step. What are the real-time computational costs associated with each algorithm?
>
> This is an important point of making the model efficient real-time. However, the real-time computational costs are a function of algorithm complexity as well as the efficient implementation. Our algorithm has an additional  $O(m^2)$ computational overhead on top of online SGD, i.e. online SGD in the final layer has cost $O(Km)$ while our method has  $O(Km+m^2)$ where K is the number of classes and m is the dimensionality of the last layer. The term $O(m^2)$ comes from a matrix multiplication needed because Kalman Filter updates a $m \times m$ covariance matrix over the weights. This is a very small additional overhead on top of the online SGD, and in fact if $K$ is much larger than $m$ the term $O(Km)$ dominates  and the cost of online SGD and our method is roughly the same. We do not aim to provide an efficient implementation of our method for real-time applications, but this could be easily incorporated into an efficient real-time implementation of Online SGD.

---

### Official Review · Reviewer_NkkH · 2023-10-30

**Soundness:** 4 excellent
**Presentation:** 4 excellent
**Contribution:** 3 good
**Rating:** 6
**Confidence:** 4

**Summary:**

The authors present an application of online Kalman filter inference within dealing with concept drift in deep learning. There is no theoretical analysis, but the computational results on variously batched and randomized classification datasets (CIFAR-100) and CLOC are promising.

**Strengths:**

The experiments show a clear improvement over a number of methods that could be seen as the state of the art, incl:
No backbone finetuning (Purely linear model)
Backbone finetuning
Backbone finetuning with Replay
Online SGD
Online SGD + Replay
ER++ (Ghunaim et al., 2023).

The paper is well written, both interms of derivation of the method and explaining the expertiments.

**Weaknesses:**

The improvement over ER++ is marginal.

The experiments do not test against the best possible Kalman filter (in terms of the system matrices), or some restarted version of the best possible Kalman filter.

**Questions:**

Have you considered extending the work of Kozdoba et al (https://doi.org/10.1609/aaai.v33i01.33014098) showing that Kalman filters can be approximated arbitrarily closely using ARMA models? This could reduce the runtime of O(m^2) to O(d) for recursion depth of d.

How exactly do you envision to use "reparametrize the integral to be an expectation under the standard normal and then apply Monte Carlo"? Plausibly, the scaleability of Monte Carlo could be a constraint? Could you present the statistical performance and runtime of the precomputing and the actual application of the method parametrized by S, as in the O(KS) or O(Km + m^2)?

---

> ### Author Response · Authors · 2023-11-22
> **Response to NkkH**
>
> We would like to thank reviewer NkkH for feedback. Please find our detailed response below.
>
> > The improvement over ER++ is marginal.
>
> We would like to highlight that we compared our method to ER++ in two scenarios: stationary CIFAR-100 and CLOC (highly non-stationary classification problem). We saw marginal improvement over ER++ in stationary CIFAR-100, see Table 1. In fact, we didn't see much performance difference across methods. In the text we argued that this is due to the fact that this problem is stationary, therefore there is not much benefit in modeling non-stationarity. After that, we compared our method to ER++ on CLOC, see Figure 3. From this Figure we can see that our method offers a significant improvement over ER++. We hope this clarifies that our method is indeed significantly better than ER++ in the non-stationary regime that we most care about.
>
> > The experiments do not test against the best possible Kalman filter (in terms of the system matrices), or some restarted version of the best possible Kalman filter.
>
> We are not sure what the reviewer means by "best possible Kalman filter". Please note that our paper is not about finding the best possible Kalman Filter nor to do Kalman Filter (KF) research. Our contribution can be summarized as follows. We propose to certain type  of KF on the last layer of the Neural Network, that crucially is fully scalable in the large scale classification application and it allows modeling non-stationarity through online learning of single scalar parameter $\gamma$.  We propose a procedure to learn non-stationarity parameters online as well as to finetune / learn the Neural Network backbone. Finally, we provide extensive empirical evidence of the effectiveness of our approach in practice.  In contrast to our scalable KF method, other more standard KF methods having more complex systems matrices will not be scalable in our application since typically will have quadratic cost wrt the number of outputs/classes; see also responses to reviewers  BEuY and VM8e.
>
>
> > Have you considered extending the work of Kozdoba et al (https://doi.org/10.1609/aaai.v33i01.33014098) showing that Kalman filters can be approximated arbitrarily closely using ARMA models? This could reduce the runtime of O(m^2) to O(d) for recursion depth of d.
>
> We would like to thank you for suggesting this paper. It is not yet clear for us how exactly to extend this work to our case. The setting of the suggested papers studies the special class of linear models - ARMA models. In our paper, we work with Neural Networks and use the Kalman Filter in the last layer. On top of that we learn the dynamics parameters (gamma) online and finetune Neural Network representation. This is not entirely clear for us how to connect our paper to this suggested paper. The scope of our paper is not about Kalman Filter research, but rather of using Kalman Filter tools in the large scale non-stationary classification problem.
>
> > How exactly do you envision to use "reparametrize the integral to be an expectation under the standard normal and then apply Monte Carlo"? Plausibly, the scaleability of Monte Carlo could be a constraint? Could you present the statistical performance and runtime of the precomputing and the actual application of the method parametrized by S, as in the O(KS) or O(Km + m^2)?
>
> As we have explained in the paper, see equation 11, the Monte-Carlo approximation is done over 1 dimensional scalars (which are phi(x)^t w) per class. Therefore the MC operations are very cheap. In practice, we found that S = 50 worked very well in all the considered settings. In practice, we found little difference in performance using smaller values for S, and because these operations are very cheap, we are not compute bound and could use even such a high S as 50.

---

### Official Review · Reviewer_reom · 2023-11-05

**Soundness:** 4 excellent
**Presentation:** 3 good
**Contribution:** 3 good
**Rating:** 6
**Confidence:** 4

**Summary:**

This paper proposes an approach to the online learning of regression and classification models using Kalman filtering. The key idea of the paper is to divide the learning into two parts: that of the feature extractor "backbone" and the "last-layer" representation. The backbone is assumed to be either pretrained and fixed, pretrained and fine-tuned, or trained from scratch using standard optimization approaches. When the backbone is fine-tuned or trained from scratch, the negative log-predictive probability is used as the objective and parameter updates are done using standard SGD steps. The last-layer representation is updated using efficient Kalman filter updates with standard Gaussian assumptions on the emission and transition densities. By allowing the weights on the last-layer representation to evolve as a Gaussian process, with a tuning parameter governing model plasticity (this parameter is shown to be learnable online), the model is able to adapt to "natural" (i.e., realistic) distribution shift. It is shown that the proposed hybrid scheme of an SGD-trained feature extractor and Kalman filter updates on the last layer allows the creation of an accurate online classifier in some large sequentially presented data sets.

**Strengths:**

Online training of neural networks with Kalman filters is not a new idea, but one that has gotten some attention recently with developments aimed at making the procedures more scalable. The present paper makes a good potential addition to this literature by proposing a method to achieve this on practically-sized neural networks for image classification problems. The paper is well-written with comprehensive appendices motivating the various design decisions made. A major strength of the paper are the convincing experimental results and carefully done comparisons. The combination of pre-training (or SGD training) with some plasticity on the parameters on the last-layer representation is, to the best of my knowledge, a novel idea. The use of a Kalman filter on the last layer only allows for efficient updates, as the number of parameters on the last layer is small relative to the rest of the network.

**Weaknesses:**

The paper is primarily focused on learning in the case of having a pre-trained extractor of a representation. Training a representation extractor can require considerable offline computational expenses. Also, having a fixed representation extractor does not ensure plasticity of the weights of the extractor itself, which may be important in cases where there is sufficient heterogeneity in the data over time that this becomes an issue. It is unclear how much of an issue this can be in practice, and therefore a comment, or some justification, should be made on the implicit assumption that plasticity in the last layer is sufficient.

As presented, the methodology does not allow for a simple incorporation of additional weight blocks into the Kalman filter update, as this breaks the linear Gaussian assumption that the method relies on. This can perhaps be circumvented with a linearization of the observation model. Have the authors considered extensions of their method in this direction? This would lead to the additional question of what weights to learn with SGD and which weights to learn with the Kalman filter (intuitively, the ones requiring the most plasticity), and how to select them.

**Questions:**

Figure 3 is perhaps the most interesting of the paper, and raises the main question that I have: why is the gap between using a pretrained extractor and Kalman filtering and SGD so much greater than between an extractor trained from scratch with Kalman filtering and SGD? Memory costs aside, why does the Kalman filter add no additional performance gain in this situation? I don't expect it to beat or match the pretrained case, of course, but perhaps be in between the SGD and the pretrained case. What is it about the interaction between learning these two parameter sets that causes this? Have the authors tried experimenting to try and improve the from scratch performance of their proposed method to be even a bit better than online SGD? An understanding of this is basically the missing part of the paper for me.

**Details Of Ethics Concerns:**

None.

---

> ### Author Response · Authors · 2023-11-22
> **Response to reom: part 1**
>
> We would like to thank the reviewer reom for the feedback. Please find our detailed answer below.
>
> > The paper is primarily focused on learning in the case of having a pre-trained extractor of a representation. Training a representation extractor can require considerable offline computational expenses. Also, having a fixed representation extractor does not ensure plasticity of the weights of the extractor itself, which may be important in cases where there is sufficient heterogeneity in the data over time that this becomes an issue. It is unclear how much of an issue this can be in practice, and therefore a comment, or some justification, should be made on the implicit assumption that plasticity in the last layer is sufficient.
>
> You are absolutely right, having non-stationarity modeled only in the last layer is more restrictive compared to modeling it in all the layers. If we use pre-trained frozen representation and the distribution shift is significant, our approach might fail to capture non-stationarity. The fact that we can update both, representation and the parameter gamma, allows us to be more flexible and capture larger classes of non-stationarity. However, we believe that this only partially mitigates the issue with plasticity. If we were to use this method in settings with extreme plasticity loss, then it is likely that our approach may not work well and more would need to be done with neural network representation. However, we are not framing our method as the solution to the plasticity problem but rather as a practical algorithm to learn in non-stationary classification domains, and in the scenarios we studied the method, it seems to work reasonably well. It would be interesting to study the performance of the method in settings with significant plasticity loss. That's said, we will add a comment about that in the text and we thank you for this useful suggestion.
>
> > As presented, the methodology does not allow for a simple incorporation of additional weight blocks into the Kalman filter update, as this breaks the linear Gaussian assumption that the method relies on. This can perhaps be circumvented with a linearization of the observation model. Have the authors considered extensions of their method in this direction? This would lead to the additional question of what weights to learn with SGD and which weights to learn with the Kalman filter (intuitively, the ones requiring the most plasticity), and how to select them.
>
> This is a very interesting suggestion. We did think about linearisation of the observation model. In fact, the paper Chang et al. does exactly that. The drawback of their approach is that they lose scalability (their complexity scales as O(P C^2), where P is the number of NN parameters and C is the number of classes) as well as that it introduces additional approximation error due to linearisation. However, what you suggest goes beyond a simple Extended Kalman Filter (EKF) approach. In fact, what you suggest is to have some set of parameters to be used for EKF (if it's not only the last layer) and the other to be used with a normal SGD. It is unclear for us how to best select such subsets and whether it could be done in a tractable  and automatic way, but this gives us a great food for thought. We will think about how to do that in the future work. Nevertheless, we will add a comment about it in the paper.

---

> > ### Author Response · Authors · 2023-11-22
> > **Response to reom: part 2**
> >
> > > Figure 3 is perhaps the most interesting of the paper, and raises the main question that I have: why is the gap between using a pretrained extractor and Kalman filtering and SGD so much greater than between an extractor trained from scratch with Kalman filtering and SGD? Memory costs aside, why does the Kalman filter add no additional performance gain in this situation? I don't expect it to beat or match the pretrained case, of course, but perhaps be in between the SGD and the pretrained case. What is it about the interaction between learning these two parameter sets that causes this? Have the authors tried experimenting to try and improve the from scratch performance of their proposed method to be even a bit better than online SGD? An understanding of this is basically the missing part of the paper for me.
> >
> > This is also a very good observation. Just to be fair to our method, it is much better than Online SGD, but it has similar performance to Online SGD with Replay. Moreover, if you check our Appendix F.5, the replay strategy in Online SGD is taken from Bornschein et al. (2022b). This is the exact extract from the Appendix "The case of online SGD with replay corresponds to using 8 replay streams on top of the learning stream which make sure that the data distribution in the replay buffer is well behaved (see Bornschein et al. (2022b) for more details)." The replay strategy the authors use makes sure to sample the previous data uniformly. Such an algorithm could be viewed as a way of implementing the Follow Regularized Leader strategy, a well known algorithm from online convex optimization. Therefore, it is not surprising that this works very well.
> >
> > For the fixed backbone and pretrained cases we believe that the large performance gap of our method versus Online SGD is due to the high data efficiency of the exact Bayesian Kalman Filter updates. The easiest case to analyze is the data stationary and fixed backbone case, where $\gamma=1$ and the Kalman Filter reduces to the online Bayesian linear regression which provides a data exchangeable and fully data efficient learning procedure (in the sense that the full information of previous data needed for prediction is stored in the posterior distribution over parameters so that the order of previous data does not matter). In contrast, the Online SGD learning rule is less data efficient since even in this stationary and fixed backbone case Online SGD is not “exchangeable” so that data order matters and affects the parameters obtained at each step.  We believe that for the “backbone finetuning from scratch case” the performance gap becomes smaller because the data efficient Bayesian Kalman Filter updates do not really help much at the beginning of training when the backbone parameters are random. Specifically, at the beginning of training, when we start from scratch and the backbone parameters are random, KF has statistics that are too noisy  which can cause learning to be slow at the first iterations. However, as mentioned above even when learning the backbone network from scratch the Kalman Filter method remains significantly better than Online SGD without replay, and only when using data replay the Online SGD becomes similar to Kalman Filter.

---

### Official Review · Reviewer_BEuY · 2023-11-07

**Soundness:** 3 good
**Presentation:** 3 good
**Contribution:** 3 good
**Rating:** 6
**Confidence:** 3

**Summary:**

The article presents a method for adapting a neural network model to a possibly non-stationary stream of data. The method consists of applying a Kalman filter to infer the weights of the last layer of the neural network over a data stream defined by an Online Continual Learning task. In addition, the neural network providing the representation and a parameter setting the level of forgetting in the last layer are adapted online, further adapting to a new problem setting. The method was applied to a simple regression task and two large-scale classification problems in continual learning which used the CIFAR-100 and CLOC datasets.

**Strengths:**

- Impressive experimental results in terms of accuracy metrics. The classification experiments demonstrate that the method can be applied to large-scale high-dimensional tasks and provides clear improvements over previous work.
- A novel method which clearly differentiates itself from similar recent work through improvements in efficiency and allowing for online adaptation of forgetting parameters
- The modeling choices were well-motivated, and the discussion in the Related work section on a meaningful setting for continual learning tasks was convincing
- Section 2.1. was well-written, as it thoroughly explains the details of online learning in the last layer, while also motivating the modeling choices

**Weaknesses:**

- It was not clear while reading the Related work section how using Kalman Filters in the representation space impacted the results compared to EKF on the full network as in Chang et al. I understood that the method presented here is more efficient, but how is the performance of the model in CL tasks impacted by only considering the last layer in the filter, especially when the representation network is trained from scratch?
- A further explanation of the backbone finetuning in the main text would be beneficial to understanding the Experiments section, since the impact of finetuning the representation neural network $\phi$ was often even greater than the impact of tuning the forgetting parameter $\gamma$, but the former did not appear in the list of contributions and was explained very quickly in the main text.
- A code implementation was not provided within the supplementary material
- Claims of efficiency in terms of computation were not supported by empirical results. The comparison to Chang et al. is supported by a theoretical analysis, but a comparison to the baselines Online SGD and ER++ would further support the claims of the method achieving improvements in accuracy metrics without sacrificing computational efficiency, especially when the backbone neural network is finetuned.
- Some visualizations could be improved for better readability. For instance, Figure 2 has very thin lines in the legend and plot which makes it difficult to see what is happening.

Chang, P. G., Durán-Martín, G., Shestopaloff, A. Y., Jones, M., and Murphy, K. (2023). Low-rank extended kalman filtering for online learning of neural networks from streaming data.

**Questions:**

How does the training of the representation neural network (backbone finetuning) impact the behaviour of the Kalman filter, since the latent space is changed mid-stream? Can this impact or explain the results in Figure 3, where the Kalman filter method performs better compared to previous work when used on a pre-trained network rather than from scratch?

---

> ### Author Response · Authors · 2023-11-22
> **Response to BEuY: part 1**
>
> We would like to thank reviewer BEuY for the feedback. Please find our detailed response below.
>
> > [...] not clear while reading the Related work section how using Kalman Filters in the representation space impacted the results compared to EKF[...], especially when the representation network is trained from scratch?
>
> Our method (last-layer KF) and EKF are quite different. EKF linearises the model likelihood $p(y|x,\theta)$ wrt $\theta$ around the previous estimate $\theta_t$, and then uses the KF to update mean and variance for the distribution of $\theta$. These parameters represent all the parameters of the Neural Network (NN). In principle, the EKF approach could represent a richer non-stationarity structure because it models it in all the NN parameters. On the other hand, as a drawback, the approximation induced by the linearisation might be problematic. The second drawback is that the naive version of EKF scales $O(P^3)$ where P represents the total number of NN parameters. Chang et al. present a low-rank plus diagonal approximation for the precision matrix which renders the complexity of the algorithm to be $O(PC^2)$, where C is the number of classes. Therefore, even using this approximation becomes problematic when applying this technique to very large-scale classification problems. Moreover, EKF replaces gradient descent on the NN parameters by KF updates applied to the linearised model. This makes it hard to combine EKF with many optimization algorithms developed for NNs. Our method uses KF only in the last layer and then updates the representation using gradient descent on parameters through backpropagation. Since it is only the last layer, this method in principle could model a limited set of non-stationarities. The advantage of the method is that it is very simple, it does not require linearizing the system and it could be used together with many powerful algorithms for NN optimization. Moreover, as we show in the paper, our method allows us to learn the parameter controlling the drift (parameter gamma) in the last layer online, allowing the system to adapt to the non-stationarity in the data distribution. This makes our approach quite flexible. In many practical problems, it might be enough to model non-stationarity in the last layer only, but in order to conclude, an additional empirical study is required.
> Finally, there exists a number of works which have shown that it is not necessary to reap the benefits of Bayesian Inference in NNs to be Bayesian wrt all parameters, and often just being Bayesian with the last layer works surprisingly well. Some papers are https://proceedings.mlr.press/v206/sharma23a.html,  https://proceedings.mlr.press/v139/daxberger21a.html, https://arxiv.org/abs/1502.05700
>
> > further explanation of the backbone finetuning in the main text would be beneficial...
>
> Thank you for the suggestion, we will add more information in the experiments section about how we finetune the backbone. In fact, we already have some information about it in the text. In the main contributions in the introduction, we point out that finetuning backbone and showing that it works empirically, is one of the contributions. Moreover, we have Appendix C where we describe in more details how we finetune the backbone. We already have references to Appendix C in text –  in Section 2.1,(ii); in Algorithm 1. We also have Appendix F which contains additional information about finetuning backbone. The reason why we use Appendix is the space constraints. Following your suggestion, we will add a reference to Appendix C in the experiments section as well as a sentence about the impact of fine-tuning backbone.
>
> > A code implementation was not provided...
>
> Unfortunately, we do not have an open-source code for this project. As an intermediate solution, we will write a more detailed pseudo-code in python format in Appendix, such that it is easier to reproduce.
>
> > Some visualizations could be improved for better readability...
>
> Thank you for this suggestion. We will improve the visibility of Figure 2.

---

> > ### Author Response · Authors · 2023-11-22
> > **Response to BEuY: part 2**
> >
> > > Claims of efficiency in terms of computation were not supported by empirical results[...], especially when the backbone neural network is finetuned.
> >
> > Online SGD on the final layer costs $O(Km)$ (i.e. for computing analytically the gradients over last layer weights) while KF costs $O(Km+m^2)$,  i.e. O(Km)  for updating the posterior Gaussian mean and $O(m^2)$ for updating the covariance matrix, where K is the number of classes and m is the dimensionality of the last layer. So essentially KF has an additional $O(m^2)$ cost due to the need to update an $m \times m$ covariance matrix, and for large number of classes K when $O(Km)$ dominates there is no noticeable computational difference between the methods.  When finetuning the backbone then we need one backpropagation step for both methods and the cost of KF and Online SGD for this step is the same. Finally, we pay an additional cost when we finetune gamma, which requires taking the gradient over a single scalar parameter. This cost can be amortized together with backpropagation for updating the backbone. Therefore, the overall cost of our method is $O(Km+m^2+B)$, where B is the cost of backpropagation. ER++ pays an additional cost of backpropagation on the replay data as well as the additional memory cost required to store the data in the replay buffer. Given the backbone network our method has an additional memory overhead of $O(Km+m^2)$,  i.e. it requires storing a K x m matrix for the posterior means and an m x m matrix for storing the shared-across-classes covariance matrix for the last layer parameters. Online SGD just needs to store an extra K x m matrix for the last layer weights, so KF just has only an extra $O(m^2)$ storage overhead compared to Online SGD without replay. Therefore, overall our method has only a minor extra overhead compared to Online-SGD or ER++, and at the same time it has performance and data efficiency benefits. At prediction, the cost of our method is equivalent to Online SGD.

---

> > > ### Comment · Reviewer_BEuY · 2023-11-23
> > > **Brief response**
> > >
> > > Thank you for your answer. As the author-reviewer period is now coming to an end, I will go through your response in more detail during the reviewer discussion period.

---

### Meta-Review · Area_Chair_gEMW · 2023-12-05

**Metareview:**

The reviewers appreciated several aspects of this work and agreed that the approach itself is novel and the paper is rather clear with some good experiments. The reviewers still suggest that the paper could benefit from additional comparisons with alternative Kalman filtering baselines for online continual learning. Additionally, the reviewers bring up possible experiments on validating whether the linear stochastic dynamics assumption makes sense in many real-world scenarios. This possible limitation is brought up by several reviewers.

Following the rebuttal and discussion phase, all reviewers recommend accepting this paper, even if the scores are borderline.

**Justification For Why Not Higher Score:**

The paper is borderline. I would not expect to see this as a spotlight.

**Justification For Why Not Lower Score:**

This paper is borderline and it could also lean towards rejection.

---

### Decision · Program_Chairs · 2024-01-16

Accept (poster)